# Bioactive Secondary Metabolites from an Arctic Marine-Derived Strain, *Streptomyces* sp. MNP-1, Using the OSMAC Strategy

**DOI:** 10.3390/molecules30081657

**Published:** 2025-04-08

**Authors:** Mengna Wu, Zijun Liu, Jiahui Wang, Wentao Hu, Huawei Zhang

**Affiliations:** 1School of Pharmaceutical Sciences, Zhejiang University of Technology, Hangzhou 310014, China; 211123070049@zjut.edu.cn (M.W.); 17857693513@163.com (Z.L.); 211123070027@zjut.edu.cn (J.W.); 2College of Pharmaceutical Science & Collaborative Innovation Center of Yangtze River Delta Region Green Pharmaceuticals, Zhejiang University of Technology, Hangzhou 310014, China; liq2363@gmail.com

**Keywords:** extremophile, *Streptomyces*, secondary metabolite, OSMAC strategy, antimicrobial activity, anticancer

## Abstract

An Arctic marine-derived strain, MNP-1, was characterized by a combined methodological approach, incorporating a variety of analytical techniques including morphological features, biochemical characteristics, and 16S ribosomal RNA (rRNA) sequence analysis. The chemical investigation of *Streptomyces* sp. MNP-1 using the OSMAC (one strain many compounds) strategy yielded the isolation of twenty known compounds (**1–20**), which were unambiguously identified by various spectroscopic approaches including ^1^H and ^13^C NMR and ESI-MS (previously reported data). Bioassay results indicated that compounds **2**, **3**, **5**, **9**, **14**, **15**, and **20** had antimicrobial activity against human pathogenic strains including *Staphylococcus aureus*, *Escherichia coli*, and *Candida albicans* with MIC values ranging from 4 to 32 μg/mL, and compounds **3** and **14** exhibited moderate inhibitory activity on A549, MCF-7, and HepG2 tumor lines showing IC_50_ values within the range of 19.88 to 35.82 µM. These findings suggest that *Streptomyces* sp. MNP-1 is one of the prolific manufacturers of bioactive secondary metabolites with therapeutic potential.

## 1. Introduction

Extremophilic microbes in polar marine ecosystems have evolved over million years and developed various unique metabolisms to survive in inhospitable environments, such as oligotrophic conditions, low temperature, and high salinity and osmotic pressure, and have been discovered to be an unexplored treasure of natural products [1,2]. Actinomycetes, particularly *Streptomyces* sp., represent a significant source of antibiotics, with these microbes producing a diversity of β-lactam, tetracycline, macrolide, aminoglycoside, and glycopeptide antibiotics that have proven clinically effective [3]. *Streptomyces* from extreme environmental sources have the potential to produce compounds that overcome multi-drug resistance and new activities, and represent a crucial source of novel antibiotics [4]. Numerous experiments have indicated that most biosynthetic gene clusters (BGCs), which are directly involved in microbial secondary metabolite (SM) production activities, are silent under conventional circumstances [5,6,7,8]. It is fortunate that the “one strain many compounds” (OSMAC) strategy has emerged as the simplest and the most effective method to activate these cryptic BGCs [9,10]. For instance, one desert-derived strain, *Streptomyces* sp. C34, was found to make three novel compounds, chaxalactins A–C, which demonstrated high activity in fighting Gram-positive bacteria, exhibiting a minimal inhibitory concentration (MIC) of less than 1 μg/mL, by varying different culture media [11]. Three new cyclopentenone chemicals, aspergispones A–C, were identified from the sea-derived fungus *Aspergillus* sp. SCSIO 41501 by means of diversity medium cultures [12].

Herein, the chemical analysis of the Arctic marine-derived isolate MNP-1 using the OSMAC strategy resulted in the separation of twenty known chemicals (**1–20**), of which compound **11** was firstly characterized from microbes and **3** and **10** were produced by the *Streptomyces* strain for the first time. In vitro bioactivity tests suggested that compounds **3** and **10** have predominant activity on the A549, MCF-7, and HepG2 tumor cell lines (see Figure 1 below).

## 2. Results

### 2.1. Strain Classification

The Arctic ore-derived isolate MNP-1 was identified by a combination of various approaches including an analysis of morphological features, biochemical characteristics, and 16S rRNA sequence analysis. The findings indicated that its single colony was round and grayish-white, and Gram’s staining test was positive (see Figure 2c below). The 16S rRNA sequence-based phylogenetic analysis (see Figure 3, Appendix A) indicated that strain MNP-1 was the closest interspecies relative to *Streptomyces acrimycini* and *S. pratensis*, suggesting it belonged to the genus *Streptomyces.*

### 2.2. Medium Evaluation

The culture conditions of *Streptomyces* sp. MNP-1 were optimized by comparing the differences of metabolites under different culture conditions using the integrated OSMAC strategy with the triple orientation of fermentation crude extract yield, compound structure prediction based on hydrogen spectroscopy characterization and LC-MS/MS metabolomics, and metabolite activity evaluation (see Appendix A). Based on the results of the comprehensive evaluation of 18 solid/liquid media (see Appendix A), three optimal medium compositions were screened to obtain a rice solid medium supplemented with 20 mg/L CuSO_4_·5H_2_O (#2), Gauze’s Synthetic Medium No. 1 with supplemented with 50 μM 5-Aza-C (#7), and MC1 liquid medium (#16).

### 2.3. Structure Elucidation

Twenty chemicals were identified in the crude extraction of *Streptomyces* sp. MNP-1 using a combination of multiple analytical chromatographic methods (see Figure 1). As demonstrated by ^1^H-NMR spectra, ^13^C-NMR spectra, and MS analyses, in a comparative analysis with previous reports, these chemicals were definitively identified as 4-methoxypyridin-2-ol (**1**) [13], 1-phenazinecarboxylic acid (**2**) [14], 4-hydroxyphenazine-1, 6-dicarboxylic acid dimethyl ester (**3**) [15], *N*-(2-hydroxy-4-methoxyphenyl)-acetamide (**4**) [16], *N*-(2-(1H-indol-3-yl)ethyl) acetamide (**5**) [17], 4-hydroxybenzoic acid (**6**) [18], 2-(acetylamino)-3-hydroxy-benzoic acid (**7**) [19], 3-(acetylamino)-4-hydroxy-benzoic acid (**8**) [20], 4-hydroxy-3-propanamidobenzoic acid (**9**) [21], 4-hydroxy-3-[(2-methyl-1-oxopropyl) amino]-benzoic acid (**10**) [22], 9-methyl-*N*-(3-methyl-2-buten-1-yl)-9H-purin-6-amine (11) [23], *N*-acetyl-*L*-tryptophan (**12**) [24], 4-(methylamino)benzoic acid (**13**) [25], staurosporine (**14**) [26], (9*Z*)-9-heptadecenoic acid (**15**) [27], (6*S*,7*R*)-6,7-methyleneheptadecanoic acid (**16**) [28], palmitic acid (**17**) [29], phenyl acetic acid (**18**) [30], daidzein (**19**) [31], and genistein (**20**) [32].

(**1**): White powder; ^1^H-NMR (CD_3_OD) *δ*_H_: 6.83 (1H, d, *J* = 7.8, H-3), 7.55 (1H, dd, *J* = 6.6, 1.8, H-5), 7.56 (1H, d, *J* = 6.6, H-6). ^13^C-NMR (CD_3_OD) *δ*_C_: 148.6 (C-2), 113.9 (C-3), 125.1 (C-4), 115.7 (C-5), 152.2 (C-6), 56.4 (4-OCH_3_). ESI-MS (*m*/*z*): 249.2 [2M − H]^−^.

(**2**): Yellow-green crystals; ^1^H-NMR (CDCl_3_) *δ*_H_: 8.98 (1H, dd, *J* = 7.1, 1.4), 8.01 (1H, m), 8.53 (1H, dd, *J* = 8.7, 1.4), 8.28 (1H, dd, *J* = 8.5, 1.5), 8.01 (1H, m), 8.01 (1H, m), 8.34 (1H, dd, *J* = 8.5, 1.5), 15.53 (1H, s). ^13^C-NMR(CDCl_3_) *δ*_C_: 125.1 (C-1), 137.6 (C-2), 130.4 (C-3), 135.2 (C-4), 143.6 (C-4a), 144.3 (C-5a), 128.1 (C-6), 133.3 (C-7), 131.9 (C-8), 130.4 (C-9), 140.0 (C-9a), 140.2 (C-10a), 166.0 (1-COOH). ESI-MS (*m*/*z*): 247.0 [M + Na]^+^.

(**3**): Purple powder; ^1^H-NMR (CDCl_3_) *δ*_H_: 7.29 (1H, d, *J* = 8.3, H-3), 7.94 (1H, dd, *J* = 7.9, H-8), 8.41 (1H, d, *J* = 7.1, H-7), 8.54 (1H, d, *J* = 6.9, H-2), 8.57 (1H, d, *J* = 8.8, H-9), 4.07 (3H, s, H-2′), 4.10 (3H, s, H-2″). HRESI-MS (*m*/*z*): 335.07 [M + Na]^+^ (calcd. for C_16_H_12_N_2_O_5_).

(**4**): Brown powder; ^1^H-NMR (CD_3_OD) *δ*_H_: 6.46 (1H, d, *J* = 2.5, H-3), 6.33 (1H, dd, *J* = 8.6, 2.5, H-5), 7.48 (1H, d, *J* = 8.6, H-6), 2.11 (3H, s, H-3′), 3.81 (3H,s). ^13^C-NMR (CD_3_OD) *δ*_C_: 119.5 (C-1), 153.9 (C-2), 100.2 (C-3), 157.1 (C-4), 107.4 (C-5), 126.0 (C-6), 172.9 (C-2′), 23.3 (C-3′), 56.1 (4-OCH_3_). HRESI-MS (*m*/*z*): 204.06 [M + Na]^+^ (calculated for C_9_H_11_NO_3_).

(**5**): Brown powder; ^1^H-NMR (CD_3_OD, 600 MHz) *δ*_H_: 7.00 (1H, m, H-6), 7.07 (1H, s, H-2), 7.09 (1H, m, H-7), 7.33 (1H, d, *J* = 8.2, H-8), 7.55 (1H, d, *J* = 7.9, H-5), 2.94 (2H, t, *J* = 7.3, H-2′), 3.47 (2H, t, *J* = 7.4, H-1′), 1.92 (3H, s, H-5′). HRESI-MS (*m*/*z*): 225.10 [M + Na]^+^ (calculated for C_12_H_14_N_2_O).

(**6**): White powder; ^1^H-NMR (CD_3_OD) *δ*_H_: 6.81 (2H, d, *J* = 8.7, H-2, 6), 7.87 (2H, d, *J* = 8.7, H-3, 5). ^13^C-NMR (CD_3_OD) *δ*_C_: 163.0 (C-4), 123.8 (C-1), 115.9 (C-2, 6), 132.9 (C-3, 5), 163.0 (C-1). HRESI-MS (*m*/*z*): 139.04 [M + H]^+^ (calculated for C_7_H_6_O_3_).

(**7**): Brown powder; ^1^H-NMR (CD_3_OD) *δ*_H_: 7.04 (1H, d, *J* = 7.9, H-4), 6.94 (1H, dd, *J* = 8.6, 1.5, H-5), 7.54 (1H, d, *J* = 7.7, H-6), 2.25 (3H, s, H-3′). HRESI-MS (*m*/*z*): 218.04 [M + Na]^+^ (calculated for C_9_H_9_NO_4_).

(**8**): White powder; ^1^H-NMR (CD_3_OD) *δ*_H_: 6.91 (1H, d, *J* = 8.4, H-5), 7.71 (1H, dd, *J* = 8.4, 2.0, H-6), 8.40 (1H, d, *J* = 2.0), 2.20 (3H, s, H-3′). ^13^C-NMR (CD_3_OD) *δ*_C_: 122.9 (C-1), 125.5 (C-2), 126.5 (C-3), 153.7 (C-4), 116.0 (C-5), 128.4 (C-6), 171.9 (C-2′), 23.1 (C-3′), 169.5 (1-COOH). HRESI-MS (*m*/*z*): 196.06 [M + H]^+^ (calculated for C_9_H_9_NO_4_).

(**9**): Yellow powder; ^1^H-NMR (CD_3_OD) *δ*_H_: 8.41 (1H, d, *J* = 1.8), 6.93 (1H, d, *J* = 8.5), 7.72 (1H, dd, *J* = 8.3, 2.0), 2.50 (2H, q, *J* = 8.2, 7.7), 1.25 (3H, t, *J* =7.4). ^13^C-NMR (CD_3_OD) *δ*_C_: 123.9 (C-1), 125.8 (C-2), 127.0 (C-3), 153.9 (C-4), 116.5 (C-5), 128.7 (C-6), 176.0 (C-2′), 30.7 (C-3′), 10.2 (C-4′), 170.2 (1-COOH). HRESI-MS (*m*/*z*): 232.06 [M + Na]^+^ (calculated for C_10_H_11_NO_4_).

(**10**): Brown powder; ^1^H-NMR (CD_3_OD) *δ*_H_: 6.92 (1H, d, *J* = 8.5, H-5), 7.73 (1H, d, *J* = 8.4, H-6), 8.37 (1H, s, H-2), 2.79 (H, m, H-3′), 1.26 (6H, d, *J* = 7.1). ^13^C-NMR(CD_3_OD) *δ*_C_: 125.8 (C-1), 127.3 (C-2), 129.3 (C-3), 152.4 (C-4), 116.7 (C-5), 130.6 (C-6), 36.8 (C-3′), 20.0 (C-4′,5′). HRESI-MS (*m*/*z*): 246.07 [M + Na]^+^ (calculated for C_11_H_13_NO_4_).

(**11**): Amorphous yellow solid; ^1^H-NMR (CD_3_OD) *δ*_H_: 8.26 (1H, s, H-4), 8.01 (1H, s, H-8), 4.18 (2H, d, *J* = 7.5, H-2′), 5.39 (H, t, *J* = 6.9, H-3′), 3.82 (1H, s, 9-CH_3_). HRESI-MS (*m*/*z*): 218.14 [M + H]^+^ (calculated for C_11_H_15_N_5_).

(**12**): White powder; ^1^H-NMR (CD_3_OD) *δ*_H_: 7.01 (1H, ddd, *J* = 8.0, 7.0, 1.0, H-6), 7.08 (1H, s, H-2), 7.08 (1H, ddd, *J* = 8.1, 7.0, 1.2, H-7), 7.32 (1H, d, *J* = 8.2, H-8), 7.56 (1H, d, *J* = 7.9, H-5), 3.16 (1H, dd, *J* = 15.1, 8.4, H-1′), 3.34 (1H, dd, *J* = 15.2, 4.7, H-1′), 4.72 (1H, dd, *J* = 8.0, 5.2, H-2′), 1.90 (1H, s, H-5′). ^13^C-NMR (CD_3_OD) *δ*_C_: 124.3(C-2), 111.1(C-3), 128.9(C-4), 119.2(C-5), 119.8(C-6), 122.4(C-7), 112.2(C-8), 138.1(C-9), 175.2(2′-COOH), 28.5 (C-1′), 54.8 (C-2′), 173.2(C-4′), 22.4(C-5′). ESI-MS (*m*/*z*): 247.4 [M + H]^+^.

(**13**): Amorphous yellow powder; ^1^H-NMR (CD_3_OD) *δ*_H_: 7.79 (2H, m, H-2, 6), 6.56 (2H, m, H-3, 5), 2.81 (3H, s, H-2′). ^13^C-NMR (CD_3_OD) *δ*_C_: 118.8 (C-4), 132.7 (C-2,6), 111.8 (C-3,5), 155.3 (C-4), 29.9 (C-2′), 171.3 (C-2′, 1-COOH). HRESI-MS (*m*/*z*): 152.07 [M + H]^+^ (calculated for C_8_H_9_NO_2_).

(**14**): Yellow-green powder; ^1^H-NMR (DMSO-d6) *δ*_H_: 7.58 (1H, d, *J* = 8.2,H-1), 7.45 (1H, t, *J* = 7.6,H-2), 7.29 (1H, m,H-3), 9.28 (1H, d, *J* = 7.9, H-4), 8.49 (1H, s, H-6), 4.94 (2H, s, H-7), 7.96 (1H, d, *J* = 7.8, H-8), 7.29 (1H, m,H-9), 7.41 (1H, t, *J* = 7.8, H-10), 7.96 (1H, d, *J* = 8.5,H-11), 4.06 (1H, d, *J* = 3.5, H-3′), 3.26 (1H, m, H-4′), 6.70 (1H, d, *J* = 3.9, H-6′), 3.32 (3H, s,), 2.30 (3H, s,), 1.46 (3H, s). ^13^C-NMR (DMSO-d6) *δ*_C_: 172.2 (C-5), 45.4 (C-7), 29.4 (C-5′), 82.8 (C-3′), 79.9 (C-6′), 91.1 (C-2′), 50.1 (C-4′), 57.3 (3′-OCH_3_), 29.7 (2′-CH_3_), 33.3 (C-2″). ESI-MS (*m*/*z*): 467.2 [M + H]^+^.

(**15**): Yellowish oil; ^1^H-NMR (CDCl_3_) *δ*_H_: 5.34 (2H, m, H-9, 10), 1.31 (18H, m), 1.63 (2H, m), 2.01 (4H, m), 2.34 (2H, t, *J* = 7.5), 0.88 (3H, t, *J* = 6.9, H-17). ^13^C-NMR (CDCl_3_) *δ*_C_: 130.2(C-9), 129.9(C-10), 180.2(C-1), 14.2 (C-17). HRESI-MS (*m*/*z*): 267.20 [M − H]^−^ (calculated for C_17_H_32_O_2_).

(**16**): Colorless oil; ^1^H-NMR (CDCl_3_) *δ*_H_: -0.33 (1H, ddd, *J* = 5.1, 5.1, 4.6, H-18), 0.56 (1H, m, H-18), 0.65 (2H, m, H-6, 7), 1.13 (2H, m), 1.32 (20H, m), 1.63 (2H, m), 2.34 (2H, t, *J* = 7.5), 0.89 (3H, t, *J* = 7.0), 1.32 (1H, m, 1-OH). ^13^C-NMR (CDCl_3_) *δ*_C_: 11.1 (C-18), 15.9 (C-6, 7), 180.1 (C-1), 14.2 (C-17). HRESI-MS (*m*/*z*): 283.26 [M + H]^+^ (calculated for C_18_H_34_O_2_).

(**17**): White flaky solid; ^1^H-NMR (CDCl_3_) *δ*_H_: 1.26 (24H, m), 1.63 (2H, m, H-3), 2.35 (2H, t, *J* = 7.5, H-2), 0.89 (3H, t, *J* = 7.0, H-16), 1.26 (1H, m,1-OH). ^13^C-NMR (CDCl_3_) *δ*_C_: 180.1 (C-1), 34.2 (C-2), 24.9 (C-3), 29.2 (C-4), 29.5 (C-5), 29.6 (C-6), 29.7 (C-7), 29.8 (C8-C12), 29.3 (C-13), 32.1 (C-14), 22.8 (C-15), 14.3 (C-16). HRESI-MS (*m*/*z*): 255.23 [M − H]^−^ (calculated for C_16_H_32_O_2_).

(**18**): White powder; ^1^H-NMR (CD_3_OD, 600 MHz) *δ*_H_: 7.27 (5H, m), 3.59 (2H, s, H-1′). ^13^C-NMR (CD_3_OD, 150 MHz) *δ*_C_: 41.9 (C-1′), 175.6 (1′-COOH), 136.0 (C-1), 130.3 (C-2,6), 129.4 (C-3,5), 127.9 (C-4). ESI-MS (*m*/*z*): 134.9 [M − 2H]^−^.

(**19**): Yellow solid; ^1^H-NMR (CD_3_OD) *δ*_H_: 8.06 (1H, d, *J* = 8.0, H-5), 6.94 (1H, m, H-6), 6.85 (1H, s, H-8), 7.37 (2H, d, *J* = 6.8, H-2′, 6′), 6.84 (2H, d, *J* = 7.0, H-3′, 5′), 8.13 (1H, s, H-2). ^13^C-NMR (CD_3_OD) *δ*_C_: 126.0 (C-3), 128.5 (C-5), 116.4 (C-6), 103.2 (C-8), 118.2 (C-10), 124.3 (C-1′), 131.4 (C-2′, 6′), 116.2 (C-3′, 5′), 154.7 (C-2), 164.6 (C-7), 159.8 (C-9), 158.7 (C-4′), 178.2 (C-4). ESI-MS (*m*/*z*): 277.0 [M + Na]^+^.

(**20**): Yellow solid; ^1^H-NMR (CD_3_OD) *δ*_H_: 8.04 (1H, s), 6.33 (1H, s), 6.22 (1H, s), 7.36 (2H, d, *J* = 9.0), 6.84 (2H, d, *J* = 7.0). ^13^C-NMR (CD_3_OD) *δ*_C_:154.8 (C-2), 123.3 (C-3), 182.3 (C-4), 163.9 (C-5), 100.1 (C-6), 165.9 (C-7), 94.8 (C-8), 158.8 (C-9), 106.3 (C-10), 124.7 (C-1′), 131.4 (2′, 6′), 116.3 (3′, 5′), 159.7 (C-4′). ESI-MS (*m*/*z*): 269.9 [M − H]^−^.

### 2.4. Bioactivity

Compounds **3** and **14** demonstrated significant inhibition effects with MICs of 4 μg/mL and 8 μg/mL on *Candida albicans* ATCC 1023, respectively. Compounds **2**, **15**, and **20** displayed inhibitory activity against *Escherichia coli* ATCC 25922 with MICs of 8–32 μg/mL. Compounds **2**, **3**, **9**, and **15** also demonstrated antimicrobial activity against *Staphylococcus aureus* ATCC 25923, with MIC values of 16–32 μg/mL. In addition, compounds **3** and **14** had moderate antiproliferative effects on MCF-7, HepG2, and A549 cell lines, with IC_50_ values in the range of 19.88 ± 1.65 µM to 35.82 ± 2.70 µM (see Table 1 below).

## 3. Discussion

In the OSMAC strategy, the best three media were selected, of which medium #2, with the addition of Cu^2+^, produced salt–ion stress effects, affecting the regulation of metabolic pathways, balancing the water and ionic environments, and inducing antioxidant mechanisms [33,34]. The addition of 5-azacytidine to #7 medium may activate the expression of metabolite-related genes by inhibiting DNA methyltransferases and demethylating DNA, leading to changes in secondary metabolite production through an increase or decrease in the corresponding amino acids. Similarly, the addition of enriched nutrients promotes metabolic pathways, regulates metabolic homeostasis, and affects the production of secondary metabolites by *Streptomyces* [35,36]. To ensure the validity of the results, it is imperative to assess the efficacy of these mechanisms through gene expression analysis and metabolic analysis under varied metabolic conditions.

Compound **2**, 1-phenazinecarboxylic acid (PCA), induces the generation of reactive oxygen species (ROS) and regulates the apoptotic protein pathway, which drives bacterial cell lysis and death [37,38,39]. Compound **14,** staurosporine, demonstrates significant antitumor effects by means of multi-targeting and multi-pathway processes. It has been shown to significantly inhibit a variety of protein kinases. In addition, it was found to upregulate pro-apoptotic proteins (Bax) and downregulate anti-apoptotic proteins (Bcl-2). The resulting effect of these actions is a delay in the apoptosis of tumor cells, which in turn prevents tumor cell proliferation and metastasis [40,41,42,43].

## 4. Materials and Methods

### 4.1. General Experimental Procedures

The measurement of nuclear magnetic resonance (NMR) spectra was determined by a Bruker Avance III-600 MHz NMR instrument (Bruker, Fällande, Switzerland). ^1^H NMR spectra were collected at 600 MHz; ^13^C NMR spectra were obtained at 150 MHz. Electrospray ionization mass spectrometry (ESI-MS) analysis was performed on a SCIEX X500 B QTOF mass spectrometer (Framingham, MA, USA). Column chromatography (CC) was conducted on ODS reverse phase silica gel (YMC Co., Ltd., Kyoto, Japan), silica gel (Qingdao Marine Chemical Inc., Qingdao, China), and Sephadex LH-20 (GE Healthcare, Danderyd, Sweden). High-performance liquid chromatography (HPLC) was carried out on an Essentia LC-20AT apparatus (Shimadzu Co., Ltd., Shanghai, China) equipped with analytical columns (Phenomenex Synergi Hydro-RP, Torrance, CA, USA, 250 × 4.6 mm, 4 µm, and Phenomenex Luna C18, 250 × 4.6 mm, 5 µm). All solvents were of analytical grade except for the chromatographic grade used for HPLC.

### 4.2. Biological Materials

An off-white bacterial strain MNP-1 was isolated from an Arctic ore sample (No. BT08-1, Appendix A), contributed by Mr. Yanhui Dong, Second Institute of Oceanography, Ministry of Natural Resources of China. The classification of the strain as *Streptomyces* is substantiated by both morphological characteristics and the analysis of an internal transcribed spacer (ITS) sequence-based phylogeny tree. The strains of *S. aureus* ATCC 25923, *E. coli* ATCC 25922, and *C. albicans* ATCC 10231 were obtained from Nanjing Medical University, Nanjing, China; lung (A549), breast (MCF-7), and liver (HepG2) cancer cell lines were obtained from the American Type Culture Collection, Manassas, VA, USA.

### 4.3. Fermentation and Extraction

The strain MNP-1 was activated by using the PDA medium plate, and after being incubated at a constant temperature at 30 °C for 3 days, the appropriate amount of colonies was picked and inoculated into PDB medium and incubated at 28 °C and 180 rpm in shaking flasks for 3 days to make the MNP-1 strain fermentation seed cultures. Each flask (2 L) contained 1 L of culture medium, was autoclaved for 20 min at 121 °C, and inoculated with 5% seed cultures. Each flask (2 L) contained 1 L of medium, was sterilized by autoclaving at 121 °C for 20 min, and seeded with 5% inoculum.

Two fermentation modes of solid fermentation and liquid fermentation are adopted, selecting three types of culture media, the salt-stressed rice solid medium, Gauze’s Synthetic Medium No. 1 with two epigenetic modifiers, and the nutrient-enriched liquid medium [44,45,46,47]. The fermentation solutions was separately diluted with equivalent volumes of EtOAC, facilitated by an ultrasonic device lasting 20 min. Subsequently, the combined organic phases were centrifuged (5000 rpm, 10 min) and the supernatants were concentrated to obtain the fermentation crude extracts I (25.5 g), II (4.3 g), and III (34.5 g).

### 4.4. Isolation and Purification

Crude extract I (25.5 g) obtained under medium #2 was subjected to fractionation and separated into 7 fractions (Fr.1–Fr.7, Appendix A) using a 200–300 mesh silica gel CC (CH_2_Cl_2_-CH_3_OH, 100:0-0:100, *v*/*v*) in gradient elution. The Fr.1–Fr.7 segments were subjected to HPLC analysis, UPLC-MS/MS molecular network prediction, and an evaluation of inhibitory activity to further identify the fractions for further isolation and purification. Among them, Fr.2 (0.2546 g) was separated by ODS reverse-phase CC (CH_3_OH-H_2_O, 50:0–100:0, *v*/*v*) and HPLC to obtain compounds **1** (4.2 mg, t_R_ = 11.5 min) and **2** (4.6 mg, t_R_ = 10.1 min). Fr.3 (0.2840 g) was separated into 6 fractions (Fr.3.1–Fr.3.6), of which Fr.3.2 was further purified by Sephadex LH-20 and HPLC and afforded compounds **3** (1.3 mg, t_R_ = 18.8 min), **4** (1.6 mg, t_R_ = 5.0 min), and **5** (1.6 mg, t_R_ = 14.6 min). Fr.3.4 was isolated to obtain compound **6** (1.8 mg, tR = 9.0 min) by HPLC (Phenomenex Synergi, 4 μm, 250 × 4.6 mm; CH_3_CN-0.1% formic acid, 11:89–15:85, *v*/*v*). Fr.4 was further isolated by ODS reverse-phase CC (CH_3_OH-H_2_O, 20:80–100:0, *v*/*v*) and HPLC to obtain compounds **7** (0.9 mg, t_R_ = 18.0 min), **8** (63.4 mg, t_R_ = 5.6 min), **9** (1.8 mg, t_R_ = 8.8 min), **10** (1.2 mg, t_R_ = 15.2 min), and **11** (1.2 mg, t_R_ = 4.7 min). Fr.5.7 was subjected to HPLC (Phenomenex Luna, 5 µm, 250 × 4.6 mm; 1.0 mL/min; CH_3_CN-H_2_O, 29:71, *v*/*v*) to provide compound **12** (2.0 mg, t_R_ = 12.7 min).

Crude extract II (4.3 g), separated under medium #7, was initially fragmented into 9 fractions (Fr.1–Fr.9, Appendix A) by MCI resin CC (CH_3_OH-H_2_O, 20:80–100:0, *v*/*v*). Based on the results of HPLC analysis, UPLC-MS/MS molecular network prediction and the evaluation of the inhibitory activity of Fr.1–Fr.9 segments, Fr.4 was further purified and separated using analytical CC (Agilent ZORBAX NH_2_, Santa Clara, CA, USA, 5 µm, 250 × 4.6 mm; 1.0 mL/min; 218/302 nm; 20 min; CH_3_CN-H_2_O, 95:5–92:8, *v*/*v*) to elute compound **13** (4.6 mg, t_R_ = 10.6 min). Fr.7 was subjected to Sephadex LH-20 and HPLC (Phenomenex Luna, 5 µm, 250 × 4.6 mm; 1.0 mL/min; 210/254 nm; CH_3_CN-0.1% formic acid, 65:35, *v*/*v*) to isolate compound **14** (2.1 mg, t_R_ = 11.5 min). The separation of Fr.8 was accomplished using analytical HPLC (Phenomenex Luna, 5 µm, 250 × 4.6 mm; 1.0 mL/min; 190/235 nm; 26 min; CH_3_CN-H_2_O, 78:22, *v*/*v*) to yield compounds **15** (26.7 mg, t_R_ = 14.7 min), **16** (15.4 mg, t_R_ = 21.3 min), and **17** (12.2 mg, t_R_ = 24.2 min).

Crude extract III (34.5 g), extracted under medium #16, was originally split into 6 fractions (Fr.1–Fr.6, Appendix A) by CC (Phenomenex Gemini Axia NX-C18, 10 µm, 50 × 21.2 mm; 210/254 nm; CH_3_CN-H_2_O, 10.0 mL/min). The Fr.1–Fr.6 segments were further separated and guided similarly to crude extract I. Based on the results of the antimicrobial activity assay, Fr.4 was separated using the analytical column (Phenomenex Luna, 5 µm, 250 × 4.6 mm; 1.0 mL/min; CH_3_CN-H_2_O, 29:71; 210/254 nm; 20 min) to obtain compounds **18** (58.3 mg, t_R_ = 9.3 min), **19** (6.2 mg, t_R_ = 11.1 min), and **20** (11.4 mg, t_R_ = 20.5 min), in that order (see Figure 4 below).

### 4.5. Antimicrobial Assay

Antimicrobial activity was determined by the gradient dilution method [48]. Indicator strains were selected using *S. aureus* ATCC 25923, *E. coli* ATCC 25922, and *C. albicans* ATCC 10231 [49]. A certain concentration of the sample and antibiotic positive control solution (bacteria: ampicillin sodium; fungi: amphotericin B) was prepared using DMSO as the solvent, and an equivalent volume of CH_3_OH was employed as the negative control. The seed solution of pathogenic bacteria was diluted using a blank medium so that its absorbance was 0.08 (UV detection wavelength 600 nm). Under the condition of aseptic operation, the concentration of the samples in each well was diluted in a gradient.

### 4.6. Cytotoxicity Assay

Three human tumor cell lines, A549, MCF-7, and HepG2, were selected to assess the antiproliferative effects of the purified chemicals using the MTT assay and were seeded into 96-well plates with a density of 5 × 10^3^ cells/well and maintained in a 5% CO_2_ tissue plate incubator at 37 °C for a total of 24 h [50]. Subsequently, 100 μL of 0.5 mg/mL MTT solution was placed into each well and incubated for 4 h. Cellular inhibition was calculated by enzyme labeling at 450 nm absorbance. Each experiment was performed in triplicate and the IC_50_ values of the samples were calculated by the GraphPad Prism 8.0 software. The determination of statistical significance was conducted using a one-way analysis of variance (ANOVA). (* *p* < 0.05, ** *p* < 0.01, *** *p* < 0.001). Doxorubicin was selected as the positive control while the negative control comprised the solvent DMSO.

## 5. Conclusions

In this research, 20 compounds (**1**–**20**) were isolated and characterized from three fermented crude extracts of *Streptomyces* sp. MNP-1. Of note, phenazine analog **3** and astrosporin derivative **14** displayed moderately inhibitory effects on *S. aureus* ATCC 25923 and *C. albicans* ATCC 10231, as well as on the tumor cell lines A549, MCF-7, and HepG2. To our knowledge, the inhibitory activity of the phenazine analog **3** was first reported against both Gram-positive bacteria and various human cancer cell lines.

## Figures and Tables

**Figure 1 molecules-30-01657-f001:**
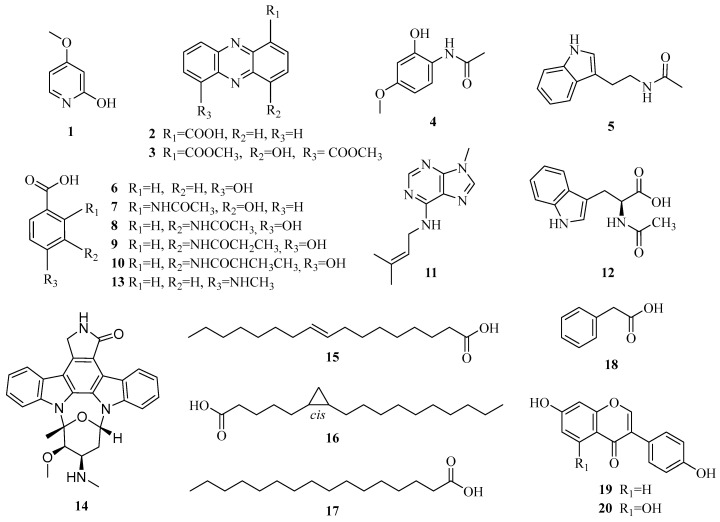
Chemical structures of compounds **1**–**20** from strain MNP-1.

**Figure 2 molecules-30-01657-f002:**
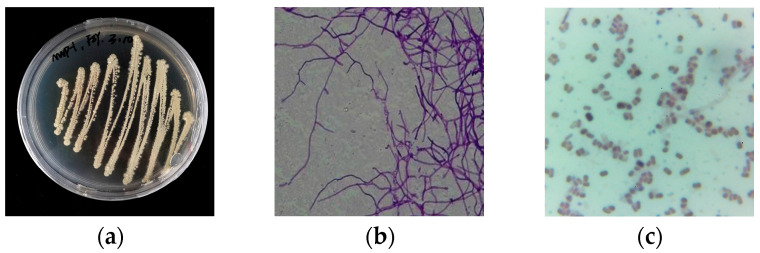
(**a**) Morphology of strain MNP-1. (**b**) Microscopic image of the mycelium and spore of the strain MNP-1 (25 × 100). (**c**) Gram staining results of the strain MNP-1 (25 × 100).

**Figure 3 molecules-30-01657-f003:**
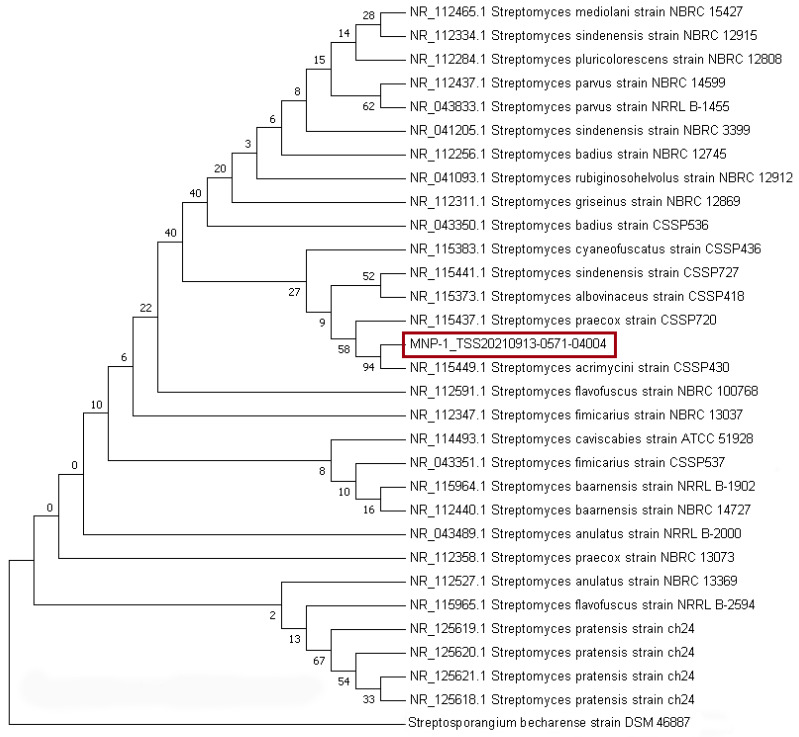
Phylogenetic tree of the strain MNP-1 (the sequence of MNP-1 is marked in red).

**Figure 4 molecules-30-01657-f004:**
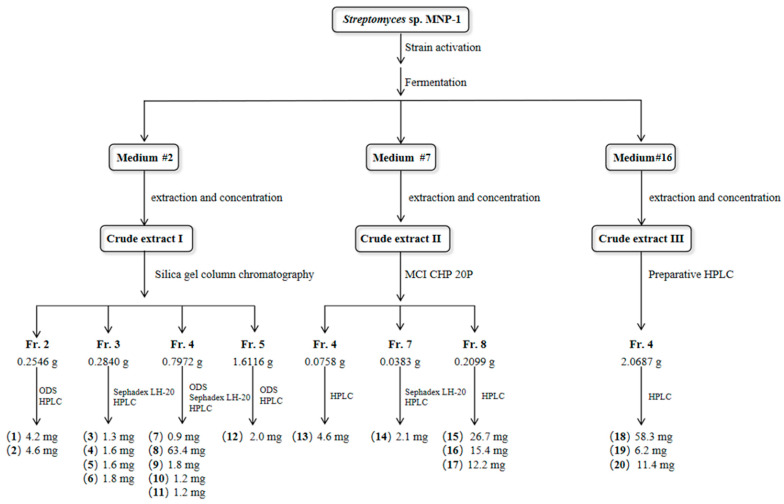
Flowchart of isolation and separation of compounds **1–20** from strain MNP-1.

**Table 1 molecules-30-01657-t001:** In vitro antimicrobial and antitumor effects of compounds **1–20**.

Compound	MIC Value (μg/mL)			IC_50_ Value (µM)		
	*S. aureus* ATCC 25923	*E. coli* ATCC 25922	*C. albicans* ATCC 10231	A549	MCF-7	HepG2
**1**	>64	>64	>64	>100	>100	>100
**2**	16	8	>64	>100	>100	>100
**3**	16	>64	4	21.52 ± 4.36	19.88 ± 1.65	35.82 ± 2.70
**4**	>64	>64	>64	>100	>100	>100
**5**	>64	>64	32	>100	>100	>100
**6**	>64	>64	>64	>100	>100	>100
**7**	>64	>64	>64	>100	>100	>100
**8**	>64	>64	>64	>100	>100	>100
**9**	32	>64	>64	>100	>100	>100
**10**	>64	>64	>64	>100	>100	>100
**11**	>64	>64	>64	>100	>100	>100
**12**	>64	>64	>64	>100	>100	>100
**13**	>64	>64	>64	>100	>100	>100
**14**	>64	>64	8	27.79 ± 6.70	35.57 ± 2.84	23.71 ± 2.89
**15**	32	32	>64	>100	>100	>100
**16**	>64	>64	>64	>100	>100	>100
**17**	>64	>64	>64	>100	>100	>100
**18**	>64	>64	>64	>100	>100	>100
**19**	>64	>64	>64	>100	>100	>100
**20**	>64	16	16	90.37 ± 2.46	>100	>100
Positive Control	0.25	1	0.25	14.86 ± 0.00	12.34 ± 0.01	15.30 ± 0.01
Negative Control	-	-	-	-	-	-

## Data Availability

Data are contained within the article and Appendix A.

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
