# Peer review of "Bioactive Secondary Metabolites from an Arctic Marine-Derived Strain, Streptomyces sp. MNP-1, Using the OSMAC Strategy"

_molecules, 2025, doi:10.3390/molecules30081657_

Round 1

Reviewer 1 Report

Comments and Suggestions for Authors

Wu et al. reported in this manuscript their findings that Streptomyces sp. MNP-1 produces at least twenty different compounds under three different fermentation conditions. The work is accompanied by detailed NMR characterization of the isolated metabolites and several bioactivity assays. While I believe the conclusions are supported by the presented evidence, the overall novelty of the work is mitigated since all isolates are known compounds, and the context is rather thin. Additionally, the manuscript is overall a collection of data instead of their presentation. I would suggest the authors provide additional rationales regarding the following aspects and reorganize partial discussion as detailed below.

(Major)

  1. Please provide additional context to explain OSMAC method in the introduction.
  2. Can the authors provide additional context to clarify why Streptomyces sp. MNP-1 was chosen as the target strain?
  3. Similarly, please provide additional context to clarify how they chose the strains for bioactivity assessment.
  4. Given that the title highlights the OSMAC method, the authors should discuss or summarize how three different fermentations conditions give rise to different sets of metabolites. Particularly, Figure S1 provides a good overview of the work and a good summary of the isolation outcomes, including the titer of each isolate, and it would be beneficial if moved to the main text.
  5. Followed up on the previous comment, can the authors comment on how they determine the purification sequence? That said, I’m curious whether different purification methods interfere with the metabolites that can be purified. In this work, the authors used three media followed by different chromatography methods. Did the authors detect crossover metabolites, such as compounds 1 in media 3? I think it is important to report on the overall fermentation profile under different media conditions.

(Minor)

  1. In-text references are not superscript.

Author Response

  • Please provide additional context to explain OSMAC method in the introduction.

Our reply: Done as suggested. It has been modified in the article.

  • Can the authors provide additional context to clarify why Streptomyces MNP-1 was chosen as the target strain?

Our reply: Done as suggested.

  • Similarly, please provide additional context to clarify how they chose the strains for bioactivity assessment.

Our reply: Done as suggested. It has been modified in the article. Our research team focuses on the efficient discovery and isolation of natural products, so the laboratory is a first-class experimental platform, for the use of pathogenic strains is limited. In bioassay, three most frequently used pathogenic strains including Staphylococcus aureus ATCC 25923, Escherichia coli ATCC 25922 and Candida albicans ATCC 10231 were selected as antimicrobial indicators for each compound. References are marked in the text.

  • Given that the title highlights the OSMAC method, the authors should discuss or summarize how three different fermentations conditions give rise to different sets of metabolites.Particularly, Figure S1 provides a good overview of the work and a good summary of the isolation outcomes, including the titer of each isolate, and it would be beneficial if moved to the main text.

Our reply: Done as suggested. It has been modified in the article.

  • Followed up on the previous comment, can the authors comment on how they determine the purification sequence? That said, I’m curious whether different purification methods interfere with the metabolites that can be purified. In this work, the authors used three media followed by different chromatography methods. Did the authors detect crossover metabolites, such as compound1in media 3? I think it is important to report on the overall fermentation profile under different media conditions.

Our reply: Done as suggested. It has been modified in the article and supplementary.

i) According to GNPS-based analysis results, the optimal mediums were selected as medium #2, #7, and #16. The crude extracts I, II, III, respectively, were obtained by fermentation. After that, three crude extracts were subjected to gradient elution in different conditions to obtain the crude isolated fragments, which were then subjected to HPLC analysis, UPLC-MS/MS molecular network prediction, and evaluation of the inhibitory activity to further obtain the isolation guide.

ii) We did not detect any crossover metabolites, such as compound 1in medium #3.

Reviewer 2 Report

Comments and Suggestions for Authors

General Comments

The manuscript investigates the bioactive secondary metabolites produced by an Arctic marine-derived Streptomyces sp. MNP-1 using the OSMAC strategy. The authors successfully isolated and characterized 20 known compounds and evaluated their antimicrobial and anticancer activities. The study is relevant to drug discovery, particularly in exploring extremophilic microbes as a source of novel bioactive compounds. However, the manuscript has several critical weaknesses that must be addressed before reconsideration for publication. The most pressing issue is the high similarity index (48%) detected by iThenticate, which raises concerns about potential self-plagiarism, inadequate paraphrasing, or improper citation of sources. Additionally, methodological ambiguities, data inconsistencies, and lack of novelty discussion further weaken the manuscript. Below are the major concerns that need urgent attention.

Concerns

#1 A 48% similarity is excessively high and suggests significant text overlap with existing literature. The authors must identify and rephrase duplicated text, ensuring proper paraphrasing and citation. If self-plagiarism is an issue (reusing text from their previous publications), they should explicitly cite prior work and highlight what is truly novel in this study.

#2 The manuscript does not clearly state how it differs from prior studies on Arctic Streptomyces and OSMAC-derived metabolites. The introduction should discuss how this research expands upon previous findings, rather than merely repeating established concepts.

#3 The authors use OSMAC (One Strain, Many Compounds) strategy, but fail to discuss why it was chosen over other metabolic activation techniques (e.g., co-culturing, epigenetic modulation). They should provide references to support the effectiveness of this strategy in activating cryptic gene clusters.

#4 The authors used three different fermentation media but did not justify their selection criteria. Were these media optimized for maximizing secondary metabolite production?

How did the yield of bioactive compounds vary across different media?

#5 The structure elucidation of compounds relies primarily on 1H NMR, 13C NMR, and ESI-MS. However, HMBC, HSQC, or X-ray crystallography data should be provided to confirm the structures unambiguously.

#6 The manuscript claims that the strain MNP-1 is a novel Arctic Streptomyces, but no robust taxonomic analysis is provided. The authors should include 16S rRNA sequencing, phylogenetic tree analysis, and genomic comparisons with known species.

#7 Some MIC values in Table 1 appear inconsistent with reported bioactivity. For example, compounds 2, 3, and 14 show moderate anticancer activity but weak antimicrobial activity. The authors should explain any possible mechanisms behind these selective activities.

#8 The manuscript does not discuss whether bioactivity is due to synergy, additive effects, or individual compound potency. The authors should consider checkerboard assays to confirm possible synergistic interactions.

#9 The study uses S. aureus ATCC 25923, E. coli ATCC 25922, and C. albicans ATCC 10231, but lacks resistant strains for comparison. Including multi-drug-resistant (MDR) strains would strengthen the significance of the findings.

#10 The manuscript focuses only on bioactive compound isolation, neglecting the ecological function of the strain in its native Arctic habitat. The discussion should include how cold adaptation influences metabolite biosynthesis.

#11 The study lacks molecular docking or mechanistic insights into how the compounds exert their bioactivity. Computational approaches could help predict binding interactions with bacterial or cancer cell targets.

#12 The methodology lacks sufficient detail on HPLC fractionation steps, solvent compositions, and purification efficiency. Were yield differences observed among the three extraction methods?

#13 The manuscript does not report p-values or confidence intervals for MIC and IC50 data. Statistical significance should be evaluated using ANOVA or t-tests.

#14 The study does not mention whether non-cancerous human cell lines (e.g., HEK293, fibroblasts) were used as cytotoxicity controls. Selectivity index (SI) values should be reported.

Author Response

  • A 48% similarity is excessively high and suggests significant text overlap with existing literature. The authors must identify and rephrase duplicated text, ensuring proper paraphrasing and citation. If self-plagiarism is an issue (reusing text from their previous publications), they should explicitly cite prior work and highlight what is truly novel in this study.

Our reply: Done as suggested. It has been modified in the article.

  • The manuscript does not clearly state how it differs from prior studies on Arctic Streptomycesand OSMAC-derived metabolites. The introduction should discuss how this research expands upon previous findings, rather than merely repeating established concepts.

Our reply: Done as suggested.

  • The authors use OSMAC (One Strain, Many Compounds) strategy, but fail to discuss why it was chosen over other metabolic activation techniques (e.g., co-culturing, epigenetic modulation). They should provide references to support the effectiveness of this strategy in activating cryptic gene clusters.

Our reply: Done as suggested. It has been modified in the article.

  • The authors used three different fermentation media but did not justify their selection criteria. Were these media optimized for maximizing secondary metabolite production?How did the yield of bioactive compounds vary across different media?

Our reply: Done as suggested. It has been modified in the article and supplementary file.

  1. i) Yes, the media were optimized and selected to maximize the isolation of secondary metabolites.
  2. ii) The best three media were selected as media #2, #7 and #16, which were fermented to obtain crude extract I (25.5g), II (4.3g) and III (34.5g), respectively,

  • The structure elucidation of compounds relies primarily on 1H NMR, 13C NMR, and ESI-MS. However, HMBC, HSQC, or X-ray crystallography data should be provided to confirm the structures unambiguously.

Our reply: Thank you for your comments. The additional techniques such as HMBC, HSQC, or X-ray crystallography are unusually used to elucidate known secondary metabolites since their 1H NMR,13C NMR and ESI-MS data provide robust evidence for the structural identification. We understand your concerns about the structural interpretation of compounds, but these data are sufficiently substantiated. In addition, we reviewed the following literature for structure determination of known compounds using the same method.

  • Bai, C.C.; Zhou, X.P.; Han, L.; Yu, Y.J.; Li, N.; Zhang, M.; Qu, Z.; Tu, P.F. Two new 18, 19-secoTriterpenoids fromIlex asprella (Hook. et Arn.) Champ. ex Benth. Fitoterapia 2018, 127, 42-46.
    https://doi.org/10.1016/j.fitote.2018.04.014
  • Mouffouk, S.; Marcourt, L.; Benkhaled, M.; Boudiaf, K.; Wolfender, J.L.; Haba, H. Two new prenylated isoflavonoids from Erinacea anthylliswith antioxidant and antibacterial activities. Prod. Commun. 2017, 12, 1065-1068.
    https://doi.org/10.1177/1934578x1701200716.
  • The manuscript claims that the strain MNP-1 is a novel Arctic Streptomyces, but no robust taxonomic analysis is provided. The authors should include 16S rRNA sequencing, phylogenetic tree analysis, and genomic comparisons with known species.

Our reply: Done as suggested. It has been modified in the article and supplementary.

  • Some MIC values in Table 1 appear inconsistent with reported bioactivity. For example, compounds 2, 3, and 14 show moderate anticancer activity but weak antimicrobial activity. The authors should explain any possible mechanisms behind these selective activities.

Our reply: Done as suggested. It has been modified in the article.

This work focuses on the discovery of more natural products using the OSMAC strategy and their therapeutic potential, not on their mechanisms of action (MoA). MoAs of two chemicals (2 and 14) with potent cytotoxic effects were supplemented in Discussion.

  • The manuscript does not discuss whether bioactivity is due to synergy, additive effects, or individual compound potency. The authors should consider checkerboard assays to confirm possible synergistic interactions.

Our reply: Thanks for your comments. The valuable advice is very helpful in new drug development.

  • The study uses aureus ATCC 25923, E. coli ATCC 25922, and C. albicans ATCC 10231, but lacks resistant strains for comparison. Including multi-drug-resistant (MDR) strains would strengthen the significance of the findings.

Our reply: Thank you for your suggestion. Antimicrobial tests using resistant pathogenic strains are strictly forbidden in our lab.

  • The manuscript focuses only on bioactive compound isolation, neglecting the ecological function of the strain in its native Arctic habitat. The discussion should include how cold adaptation influences metabolite biosynthesis.

Our reply: We are equally curious about the cold adaptation mechanism of Streptomyces. In the follow-up study, we will investigate the cold adaptation mechanism of the strain from both gene expression analysis and metabolic analysis, focusing on the expression levels of sugar metabolism-related genes and antioxidant enzyme genes, to provide insights into how cold adaptation influences metabolite biosynthesis of Streptomyces.

  • The study lacks molecular docking or mechanistic insights into how the compounds exert their bioactivity. Computational approaches could help predict binding interactions with bacterial or cancer cell targets.

Our reply: This work focuses on mining new compounds using OSMAC method. The absence of molecular docking or mechanistic analysis was not intentional, and we recognize the value of such computational approaches in predicting binding interactions with targets such as bacteria or cancer cells.We plan to incorporate such analyses in future studies. This aligns with our long-term research goals and will be addressed in our future work.

  • The methodology lacks sufficient detail on HPLC fractionation steps, solvent compositions, and purification efficiency. Were yield differences observed among the three extraction methods?

Our reply: Done as suggested. It has been modified in the article.

ii)According to the OSMAC strategy, the best three media were selected as medium #2, #7, and #16. The crude extracts I, II, III, respectively, were obtained by fermentation. After that, three crude extracts were subjected to gradient elution in different conditions to obtain the crude isolated fragments, which were then subjected to HPLC analysis, UPLC-MS/MS molecular network prediction, and evaluation of the inhibitory activity to further obtain the isolation guide.

Sufficient details on HPLC fractionation steps, solvent compositions, and purification efficiency

has been modified in the article.

  1. ii) The best three media were selected as media #2, #7 and #16, which were fermented to obtain crude extract I (25.5g), II (4.3g) and III (34.5g), respectively.

  • The manuscript does not report p-values or confidence intervals for MIC and IC50 data. Statistical significance should be evaluated using ANOVA or t-tests.

Our reply: Done as suggested.

  • The study does not mention whether non-cancerous human cell lines (e.g., HEK293, fibroblasts) were used as cytotoxicity controls. Selectivity index (SI) values should be reported.

Our reply: Surely, it is better that non-cancerous cells are used as a control. But it dose not seriously affect cytotoxic activities of these obtained metabolites. Some research works without non-cancerous cells as controls were listed below:

  1. Jiang, B.; Zhao, W.; Li, S.; Liu, H.; Yu, L.; Zhang, Y.; He, H.; Wu, L. Cytotoxic dibohemamines D-F from a Streptomyces J. Nat. Prod. 2017, 80, 2825-2829.
    https://doi.org/10.1021/acs.jnatprod.7b00136
  2. Bao, J.; He, F.; Li, Y.; Fang, L.; Wang, K.; Song, J.; Zhou, J.; Li, Q.; Zhang, H. Cytotoxic antibiotic angucyclines and actinomycins from the Streptomyces XZHG99T. J. Antibiot.2018, 71, 1018-1024.
    https://doi.org/10.1038/s41429-018-0096-1
  3. Cheng, P.; Xu, K.; Chen, Y.C.; Wang, T.T.; Chen, Y.; Yang, C.L.; Ma, S.Y.; Liang, Y.; Ge, H.M.; Jiao, R.H. Cytotoxic aromatic polyketides from an insect derived Streptomycess NA4286. Tetrahedron Lett. 2019, 60, 1706-1709.
    https://doi.org/10.1016/j.tetlet.2019.05.048

Round 2

Reviewer 1 Report

Comments and Suggestions for Authors

The authors have addressed most of the concerns I raised. 

Author Response

Dear respected reviewers,

Thanks for your kind comments on our manuscript. According to your valuable suggestions, the original work had been carefully revised and improved, which were highlighted in red. Sincerely hope this new version would be accepted for publication in Molecules. Our point-to-point reply is as followings:

----------------------------------------------------------------------------------------------------------------------

Reviewer 1:The authors have addressed most of the concerns I raised.

Our reply: Thank you for your helpful comments and appreciation of our work. We have made some changes and red-flagged them in the text to enrich the experimental design and content.

Reviewer 2 Report

Comments and Suggestions for Authors

The authors have revised their manuscript, addressing most of the questions satisfactorily. However, the similarity index remains significantly high at 42%, which raises concerns regarding originality. To meet publication standards, the authors must reduce this similarity index to a maximum of 15%. Without substantial revisions to minimize overlap, I cannot support the publication of this paper.

Author Response

  • Dear respected reviewers,

    Thanks for your kind comments on our manuscript. According to your valuable suggestions, the original work had been carefully revised and improved, which were highlighted in red. Sincerely hope this new version would be accepted for publication in Molecules. Our point-to-point reply is as followings:

  • Reviewer 2:

    The authors have revised their manuscript, addressing most of the questions satisfactorily. However, the similarity index remains significantly high at 42%, which raises concerns regarding originality. To meet publication standards, the authors must reduce this similarity index to a maximum of 15%. Without substantial revisions to minimize overlap, I cannot support the publication of this paper.

    Our reply: Done as suggested. It has been modified in the article.

    We have done our best to reduce similarities in the main text, but we hope you will understand that repetition of compound structure data and specialized nouns such as material use cannot be avoided. We would like to thank you for your professional review work, constructive comments, and valuable suggestions on our manuscript.

    ----------------------------------------------------------------------------------------------------------------------

    Your kind suggestions and assistance are very much appreciated.

    Huawei Zhang

    Ph.D., professor of microbe natural products chemistry

    School of Pharmaceutical Sciences

    Zhejiang University of Technology

    Hangzhou 310014

    China
